# Health Claims for Protein Food Supplements for Athletes—The Analysis Is in Accordance with the EFSA’s Scientific Opinion

**DOI:** 10.3390/nu17111923

**Published:** 2025-06-03

**Authors:** María Dolores Rodríguez-Hernández, José Miguel Martínez-Sanz, Carlos Javier García, José Antonio Gabaldón, Federico Ferreres, Miguel Escribano, Daniel Giménez-Monzó, Ángel Gil-Izquierdo

**Affiliations:** 1Research Group on Quality, Safety, and Bioactivity of Plant-Derived Foods, Department of Food Science and Technology, CEBAS-CSIC, Campus de Espinardo-25, 30100 Murcia, Spain; mdrh@um.es (M.D.R.-H.); cjgarcia@cebas.csic.es (C.J.G.); miguel.escribano@mecollective.es (M.E.); 2Department of Nursing, Faculty of Health Sciences, University of Alicante, 03690 Alicante, Spain; 3Molecular Recognition and Encapsulation Group (REM), Health Sciences Department, Universidad Católica de Murcia (UCAM), Campus Los Jerónimos 135, 30107 Murcia, Spain; jagabaldon@ucam.edu (J.A.G.); fferreres@ucam.edu (F.F.); 4Department of Community Nursing, Preventive Medicine and Public Health and History of Science Health, University of Alicante, 03690 Alicante, Spain; dgimenez@ua.es

**Keywords:** nutrition, sport, protein supplements, muscle mass, health claims, fraud

## Abstract

Background: Protein supplements are among the most popular, available and growing complementary products. Fraud related to the mislabeling, inaccurate analysis or declaration of ingredient quantities, and health claims not aligned with those approved by EFSA is high. This study aims to analyze the claims related to protein supplements in commercial messages. Methods: An observational cross-sectional study was conducted to analyze the content and the degree to which health claims stated on the labeling or technical data sheets of protein supplements comply with those authorized by current European legislation and supported by existing scientific evidence. The products were searched for using Amazon and Google Shopping. Results: Of the 209 health claims evaluated, 60 claims fully complied with the recommendations, representing 28.7% of the total (*n* = 209). In contrast, 12 claims in which the stated text did not conform to the health claims established by EFSA were identified, representing 5.7% of the total (*n* = 209). The most widely used unauthorized health claims on the market are those referring to “Post-workout recovery” (11.1%), followed by “Promotes muscle recovery (casein)” (9.5% each), referring to whey protein and casein, respectively. Of all the products analyzed in the study, 43.8% (*n* = 46) of the products made health claims not authorized by the EFSA. Conclusions: These findings suggest that the high-quality advertising of protein supplements should engage consumers, industry stakeholders, scientific research, and the European Food Safety Authority to ensure compliance with European regulations, provide accurate guidance for manufacturers, and protect consumer rights under current legislation.

## 1. Introduction

### 1.1. Sports Food Supplements

Dietary supplements are concentrated sources of nutrients or other substances with a nutritional or physiological effect that are marketed in “dosage” forms (e.g., pills, tablets, capsules, liquids in measured doses). They may contain a wide range of nutrients and other ingredients, such as vitamins, minerals, amino acids, essential fatty acids, fiber and various plant and herbal extracts, among others [1].

According to the International Olympic Committee (IOC), nutritional supplements for athletes or dietary supplements (SFSs) are defined as “a food, component, nutrient or non-food product, component that is intentionally consumed within a normal diet for the of obtaining a particular effect on health or performance”. These substances are consumed because of the existence of competitive situations in which athletes aim to achieve certain objectives related to SFSs [2]. They are consumed by up to 90% of athletes, depending on the sport, being a common practice in most athletes [3,4,5,6].

This consumption of sports supplements is becoming more widespread among the sports community, both elite and amateur. In recent years, their consumption has increased both internationally and nationally [7].

In parallel to the increase in the consumption of protein supplements, ergogenic aids have grown exponentially. More and more varieties of protein supplements are becoming available, offering improvements in composition and new formulations [5].

In particular, the consumption of protein products in the last few years in different types of sports is as follows: rowers (whey protein 65%), elite soccer players (whey protein 48.6%), Turkish soccer players (whey protein 28.2%), mountain runners (sports bars 81.9%), open water swimmers (whey protein 15.9%), Spanish triathletes (protein bars 40.9%, whey protein 29.7%), national squash players (whey protein 28.6%), Spanish basketball players (whey protein 29.7%), handball players (protein bars 25.6% and whey protein 30.4%) [8,9,10,11,12,13]; as it can be observed, protein products are a type of supplement present in different sports.

According to the Australian Institute of Sport (AIS), the above SSFs are within evidence group A (compatible with use in specific situations in sport, using evidence-based protocols) [14].

### 1.2. Sports Food Supplements, Scientific Evidence and Legislation

SFS consumption by athletes is influenced by specific legal frameworks and regulatory provisions [15], as well as by official guidelines and scientific work from organizations such as the AIS [14]. These regulations are intended to offer guidance on proper usage, including dosage instructions, safety profiles, possible risks, and necessary warnings [15]. Furthermore, they should also address aspects such as product accessibility and market availability, in addition to presenting evidence regarding their potential to enhance athletic performance. These elements reflect key principles in public health aimed at helping individuals achieve and sustain optimal health [16]. Despite this, certain widely used supplements, including glutamine and L-carnitine, are marketed as SFSs despite the lack of objective (scientific) evidence to support their claimed effectiveness [17].

Policies and regulatory standards concerning SFSs differ across nations and also depend on the specific category of product. Within the European Union and its member countries, multiple regulations related to SFSs are in place. These generally cover aspects such as labeling—particularly claims related to health or performance—alongside issues of safety, promotional practices, and the permitted levels of vitamins, minerals, and other active ingredients [18].

At present, the legislation related to the regulation and application of nutritional ergogenic aids or sports food products can be found in the following documents: Regulation (EU) No. 1169/2011, Regulation (EC) No. 353/2008, Regulation (EC) No. 1924/2006, Regulation (EC) No. 1925/2006, and Directive 2002/46/EC. Regulation (EU) No. 609/2013 [15]. Despite the existence of this framework, the regulation lacks a regulatory section on the use and application of SFSs by consumers [15].

Research institutions and public institutions such as the European Food Safety Authority (EFSA) have conducted evaluations on the properties and effects of the different substances added or isolated in supplements, as well as the safety of consuming of these substances, including protein supplements [19].

### 1.3. Protein Supplements

Protein supplements are one of the most popular and widely used by consumers, with their global market value projected to be USD 21.5 billion by 2025. Sports nutrition is the main application for protein supplements and the fastest growing sector is plant-based proteins [14].

Protein supplements are available as stand-alone products in powder form, bars and ready-to-drink shakes. In recent years, a growing trend has emerged in which commercially available foods are enriched with isolated proteins, such as those found in breakfast cereals or nutritional bars. A wide range of protein supplements is now available, differing in both form and source [14].

Numerous studies have demonstrated the effect of protein supplementation on muscle mass gains. Systematic reviews and meta-analyses have shown the effect of protein supplementation on resistance-training-induced muscle mass and strength gains in healthy adults, and improved changes in muscle strength and size during prolonged resistance exercise training in healthy adults [2,20]. The decision to use a protein supplement should only be made after the consideration of several factors, including the athlete’s training load and goals, lifestyle commitments, daily energy requirements, existing dietary plan, the practicalities of post-exercise scenarios, and available finances [14,21].

The EFSA [19] has issued opinions in favor of health claims related to proteins, and these can be used to provide information to the consumer. Such claims occupy a central position in the marketing of the product, both in its labeling and in advertising the product to athletes [22].

However, research in this area has identified instances in which claims on protein supplements are not compliant with regulations [23]. Substantial regulatory infractions regarding the health claims on supplements have been identified [24,25]. A recent analysis of substitute drinks showed that no health claims were compliant with the EFSA’s opinions, where the majority of claims communicated an unproven cause-and-effect relationship with the product and a health outcome [26].

t has not yet been evaluated whether the health claims used on the labels and advertisements for protein supplements in the EU market are compliant with the EFSA’s opinions and approved health claims. The objectives of this study were to identify the health claims used on protein supplements in the EU market and verify their compliance with current European regulations and the veracity of the information conveyed to the consumer.

## 2. Materials and Methods

### 2.1. Type of Study

An observational and cross-sectional study was carried out based on an analysis of the content and adequacy of the health claims indicated in the data sheet of product or company-website-based sports food supplements compared to those established by current European legislation and the evidence described to date. In addition, the design of the study, as well as the development of the manuscript, followed the STROBE statement [27].

### 2.2. Study Population Selection Strategy

The search for the sample products was conducted in October 2023 through the web shopping platforms Amazon and Google Shopping. These websites were selected because they are the main online shopping websites. The following terms were used in the search process: “proteins”, “protein”, “casein” were entered in both portals. From this initial search, supplements that were only protein supplements were selected. Subsequently, each of the web portals of the selected supplement brands were used to obtain the health claims for each supplement (see Appendix A). The process of obtaining each component of the sample was different depending on the portal visited.

### 2.3. Inclusion Criteria

In this study, the selected sample included SFSs defined as protein supplements and offered for sale in Europe. SFSs that were not defined as protein supplements, that appeared several times in the search, on the same website or both, or that did not include health-related claims were excluded.

### 2.4. Data Extraction

After conducting the search to select the study sample, a descriptive analysis of the characteristics of each product in the selected protein supplements was performed based on the technical data sheets or company websites.

The variables studied for each product in the sample were as follows:Product name: name of each of the supplements belonging to the study sample;Sportsbramd: brand name of each supplement in the sample. The SFSs in the sample were defined (Appendix A);Health claims for protein supplements: those present on the product’s technical data sheet or company website of each of the supplements in the requested sample;Dosage: amount of product consumed, as recommended by the manufacturer in each supplement;Nutritional information: value of nutrients, carbohydrates, proteins, fats and salt in each of the supplements in the sample.

### 2.5. Data Analysis

After data extraction, an analysis classifying the health claims indicated on the product data sheet or on the company’s website was carried out, according to the approved health claims. In this analysis, the following variables were obtained:Approved health claims: EFSA-approved health claims for proteins;Total supplements in which this statement is made (no. and %): total number and percentage of supplements in the sample in which this statement is made;Comply with the conditions of use: conditions based on Regulation (EC) No. 432/2012 [28];Number and percentage of supplements in which these conditions of use are met for this claim: number and percentage of supplements in the sample in which the conditions of use are met for each claim;Unauthorized health claims. Relation to health: health claims not authorized by the EFSA for whey and casein proteins;Health claims stated on the product or on the company’s website: health claims stated on each of the supplements in the sample;Number and percentage of supplements in the sample where the claim appears, either on the product or the company’s website Adequacy yes/no: whether the health claims of each of the supplements in the sample conform to the health claims defined by the EFSA;Reason: according to the EFSA’s scientific opinion, the reason for compliance or non-compliance and the proposed modification of the supplements in the sample to achieve a better adaptation to the approved health claims. Number 1 to 5, from non-compliant to compliant, respectively. Number 1. Reason: not in accordance with the approved protein claims and the recommended appropriate dosage of the product. Proposed modification: delete product claim. Number 2: Reason: Conforms to the recommended appropriate dosage of the product, but the text of the statement indicated does not conform to the approved one. Proposed modification: modify the declaration by specifying the exact text of the approved declaration Number 3. Reason: Conforms to the appropriate recommended dosage of the product, but the text of the approved statement is missing. Modification proposal: modify the statement by specifying the exact text of the approved statement. Number 4. Reason: Conforms to the appropriate recommended dosage of the product, but some words in the text of the declaration need to be changed. Proposal for amendment: amend the declaration by specifying the exact text of the approved declaration. Number 5. Reason: Conforms to all of the above. Proposed modification: do not modify or delete the declaration.

### 2.6. Compliance with Legislation and Scientific Evidence

Once the content analysis of the products in the selected sample had been completed, a comparison was made between the different health claims made for the protein supplements included in the product data sheet or company’s website, in order to determine whether they were in line with the scientific evidence established by the EFSA (Table 1).

## 3. Results

A total of 156 results were obtained in the search, of which 102 SFSs belonging to different commercial brands met the inclusion criteria established in the methodology. Nineteen were rejected because they appeared duplicated on the same web page or both, 26 because they contained ingredients other than protein, and 9 because they did not specify health claims (Figure 1). For the 102 selected SFSs, the health claims present and their dosage are specified.

### 3.1. Indicated Conditions of Use of the Product

As can be seen in Table 2, with respect to the conditions of use for each product and the statement “Proteins contribute to increase muscle mass”, 100% (*n* = 99) comply with the conditions of use. Regarding the statement “Proteins contribute to the preservation of muscle mass”, 100% (*n* = 79) comply with the conditions of use. Regarding the statement “Proteins contribute to the maintenance of bones in normal conditions”, 100% (*n* = 31) comply with the conditions of use.

### 3.2. Health Claims Not Authorized by the EFSA

Table 3 shows the percentage distribution of each EFSA-rejected health claim found in the sample of SFSs and the type of EFSA-unauthorized health claim indicated by the manufacturer of each SFS.

The most frequently described unauthorized health claims for the SFSs are “Post-workout recovery”, at 11.1%; this is followed by “Supports muscle recovery (casein)”, each at 9.5%, “Supports muscle recovery (whey protein)” at 7.9%, and “Improves recovery”, “Supports muscle growth for even better performance”, and “Improves sports performance”, representing 6.3%. This is followed at the same time by “Micellar casein perfect to avoid catabolization”, representing 4.8%, and “Whey protein benefits recovery of muscle fibers”, “Recommended for those seeking to recover their muscle tissues after training”, “Tissue recovery”, “Whey protein increases muscle volume”, “Increased strength”, “Your protein synthesis is accelerated and your body can build muscle”, and “Helps decrease appetite”, corresponding to 3.2%”.

On the other hand, there are statements that only appear on a single sports product in the sample of supplements, such as “Target: Endurance”, “Improve performance and recover quickly from workouts”, “Key amino acids for proper muscle function and recovery”, “Whey protein benefits muscle growth”, “Objective whey protein muscle development muscle maintenance and muscle recovery”, “Whey protein with amino acids helps increase muscle mass and promotes the maintenance of muscle tissue”, “Improves strength”, “Supports fat loss”, “Helps decrease appetite, calm the urge to eat”, “stimulate protein synthesis, the process that makes muscles grow because the amino acids it contains are transported to the muscles through the bloodstream”, “Satiety control”, “Weight loss”, “A premium whey protein isolate formulated to feed your muscles fast, so you can recover faster”, “Prevents muscle catabolism”, “Promotes recovery of damaged fibers versus catabolism”, and “Supplies the amino acids your body needs to recover while you sleep”. Each of these represents 1.6% of the total statements in the sample.

Regarding products that claim that casein promotes muscle recovery, they should clarify that it is not casein that contributes to muscle recovery and the supposed growth or maintenance of muscle mass, but proteins in general.

Several products that could mislead the consumer by indicating that the essential amino acids contained in whey protein help repair muscles were found, but by adding the following sentence “Protein contributes to the development and maintenance of muscle mass”, it seems that the manufacturer has used the principle of flexibility, as he has mentioned the EFSA-approved health claim.

On the other hand, products with messages such as “Highly concentrated and balanced whey protein” were also found; this is not a claim but rather information about the protein.

### 3.3. Health Claims, and Compliance with Current European Legislation and Scientific Evidence

Table 4 presents a percentage distribution of each health claim found in the sample of SFSs and the type of health claim indicated by the manufacturer of each SFS.

The health statements most frequently described in the SFSs refer to “muscle development or muscle mass (helps, favors, promotes)” and “contribute to the maintenance, maintain muscle mass, muscles”, each appearing in 10% of the cases; this is followed by “protein contributes to the maintenance of muscle mass”, “protein contributes to the growth, development of muscle mass”, and “contributes (to the growth, increase, development, gain) of muscle mass”, with 8.6%, 7.7% and 7.2%, respectively, followed in turn by the statements “muscle growth (favors, helps, supports, promotes, encourages, perfect for)” with 6.2%, “The proteins contribute to the maintenance of bones in normal conditions” with 5.3%, “protein (100% whey isolate, 100% casein, whey)” and “contributes to the maintenance of muscle mass”, with 4.8%.

On the other hand, there are statements that only appear on single sports supplement samples, such as “Promotes muscle building”, “Designed to help you build muscle”, “To accelerate muscle building”, “Protein is important for the maintenance of our muscles”, “Provides amino acids in a sustained manner so we avoid losing muscle mass”, “prevents loss of muscle mass/tone”, “promotes muscle retention”, “to feed your muscles”, “muscle definition”, “100% casein protein contributes to the maintenance of bones under normal conditions”, “Protein content contributes to the maintenance of proper bone health”, “maintenance of healthy bones”, “can help you maintain healthy bones”, and “promotes bones”. In each case, these statements contribute 0.5% of the total sample.

Table 5 establishes a percentage distribution of each nutrition claim found in the sample of SFSs and the type of nutrition claim indicated by the manufacturer on each SFS.

These products with nutritional claims have been excluded because the purpose of this study refers to health claims, not nutritional claims.

Table 6 shows the percentage distribution of the non-health related messages found in the SFSs, and which have been indicated by the manufacturers of each SFS.

Messages not related to the manufacturer’s health claims were found in the products in our study; these products were not accompanied by a specific authorized and related health claim, so these products have been excluded because they do not indicate health properties.

### 3.4. Degree of Compliance and Proposals for Modifications

The reasons for the adequacy or inadequacy of the SFS product claims analyzed, as well as the possible modifications to be made, are shown in in Table 7 based on what has been established by the scientific reference institutions (Table 1).

According to Table 7 (summary of distribution of conditions of use (doses, etc.) of products according to health claims and their adequacy), SFS products with the claims “Proteins contribute to the increase of muscle mass”, “Proteins contribute to the preservation of muscle mass”, and “Proteins contribute to the maintenance of bones under normal conditions” met the adequacy ratios of the claims, representing 25.3%, 30.4%, and 35.5%, respectively, for each type of claim.

SFS products with the statements “Proteins contribute to the increase of muscle mass”, “Proteins contribute to the maintenance of muscle mass”, and “Proteins contribute to the maintenance of bones in normal conditions” almost met the adequacy ratios of the claims representing 15%, 34.2%, and 38.7%, respectively, for each type of claim.

## 4. Discussion

In the present study, we analyzed the different health claims made for SFSs included in the technical data sheets or the company’s websites of a product sample, as well as the terms and conditions of use indicated for the achievement of these effects. In our results, we found products that met all the required conditions of use, and the degree of adequacy to the health claim was in the claim “Proteins contribute to increase muscle mass”, corresponding to 25.3% (*n* = 25) of products, in the claim “Proteins contribute to preserve muscle mass”, corresponding to 30.4% (*n* = 24) of products, and in the claim “Proteins contribute to the maintenance of bones in normal conditions”, with 35.5% (*n* = 11).

In total, 100% of the products comply with the conditions of use; however, 71.3% of the products analyzed have to modify the adequacy of the required health claim. Products with nutrition claims were found, where the claim “High protein content” accounted for 66.7% (*n* = 2) of the products.

### 4.1. Health Claims and Proposed Dosages

Athletes and people involved in sporting activities frequently consume protein supplements. However, purchasing decisions are often influenced by commercial claims rather than evidence-based research [30].

There are studies on the impact of package labeling and other sources of information on nutritional supplement purchasing decisions, where there are deficiencies in current labeling information practices and a need for practical intervention, policy development and/or regulation [31].

In another study, the claims of protein supplements marketed for sports practice were analyzed, where it was found that some whey proteins and plant protein supplements made claims of muscle mass enhancement without being authorized to do so; for example, they claimed that it accelerates the recovery process and prevents muscle catabolism [32].

This study identified and compiled the various health-related claims found across the SFS products analyzed, revealing a total of 38 distinct health effect claims made by manufacturers; of these, only six claims were in full compliance with the institutions. These approved claims have received authorization at the European level by the EFSA and the European Commission (EC) [28]. Additionally, specific legislation has been established to regulate both the formulation and marketing of sports nutrition products, supported by consensus-based documents and a registry listing the health claims permitted for use in the promotion of these items [33].

It has been noted that some products require the removal of their claim because they do not comply with either the approved protein declarations nor the recommended appropriate dose, as established by the institutions [28].

It is important to understand that sometimes the labeling, presentation and advertising of a food may include markings, symbols or general mentions (for example: “good for your skin”), which are considered general health claims that are not specifically authorized. All of these must be accompanied by a specific authorized declaration and be related to the claim [34].

It is also necessary to point out that in the marketplace, we can find claims that are not identical to those authorized at the European level. A certain flexibility in their wording is allowed, but the adapted text must have the same meaning for consumers as that used for the authorized claims, and be subject to the same conditions of use. It cannot be made stronger, become misleading, or refer to a medicinal claim, because no food can be said to prevent, cure or treat a disease, according to the General Principles of Flexibility for the wording of health claims; these are recommendations developed by experts from member states and approved in 2012 [34].

In our study, we found products that complied with the degree of adequacy of the health claims approved by the EFSA. Taking into account the General Principles of Flexibility, the claims observed are not identical to those authorized at the European level, but they have the same meaning for consumers, representing 17.2% (*n* = 36) of the products.

The EFSA is responsible for authorizing and refusing health claims, which are contained in the Community register, together with their conditions of use and the reasons for the rejection in the case of refused claims. Authorized claims have previously undergone a procedure in which they have to demonstrate that they are based on sound scientific data that are evaluated and authorized at the European level (Regulation No. 1924/2006) [29]. In our study, among the claims rejected by the EFSA, the most used on the market were those referring to “Post-training recovery”, at 11.1%. This was followed by “Promotes muscle recovery (casein)”, at 9.5%. One of these claims referred to whey protein and the other referred to casein. Of all the products analyzed in our study, 43.8% (*n* = 46) of the products named health claims not authorized by the EFSA.

Products mentioning health claims rejected by the EFSA, e.g., “whey protein produces post-workout recovery,” should remove such claims, as has been observed in other studies where health claims that did not comply with EFSA-approved health claims were detected [35,36].

### 4.2. Fraud in Advertising and Direct Consumer Information

Adulteration and fraud in food products have existed since the beginning of food processing and production. However, these fraudulent practices are becoming increasingly frequent in modern times [37]. This type of fraud has been observed in a study on health claims related to recovery beverages [25], a study on the caffeine dosage in sports supplement labeling [35], and a study on health claims regarding creatine monohydrate [36].

Considering that the major cause of unintentional doping is the inadequate and incorrect use of dietary supplements, and that the athlete is ultimately responsible according to the strict liability code of the World Anti-Doping Agency (WADA), it would be ethical for health and sport professionals to evaluate the legality, safety and efficacy of dietary supplements for athletes in order to minimize the risk of doping. It would also be sensible for their decision to be based on scientific evidence and criteria for use [24].

However, there are studies that have analyzed how consumers gather information when choosing nutritional supplements. An important finding was that coaches, physical and/or gym trainers and athletes (24%) have a greater influence on the choice of nutritional supplement products than that of pharmacists, dieticians, nutritionists and physicians (10%). Deficiencies in current labeling information practices were also found [31].

Since product labeling significantly influences consumer purchasing behavior, ensuring accurate and transparent information on supplement labels is crucial [31].

In addition, athletes can become dependent on inadequate sources of information, and may be largely unaware of means of detecting supplement contamination. Urgent nutrition education and counseling should be provided to both athletes and coaches, with a focus on the role of nutritionists and sports scientists, the acute and long-term side effects of incorrect supplementation strategies, and techniques for identifying potential supplement contamination [3].

### 4.3. Action to Be Taken in the Face of Advertising Fraud

The EFSA is involved in food safety in the context of Public Health, with regard to the advertising and marketing of food products. In addition, regulation through legislative documents serves as a legally enforceable tool against advertising and food fraud [34].

In Spain, the Spanish Agency for Food Safety and Nutrition (AESAN) operates within the competence framework of the General State Administration, carrying out functions related to food safety and public health nutrition, such as promoting and encouraging the collaboration and coordination of public administrations in matters of food safety and nutrition; this is particularly related to its role in communicating with the European Food Safety Authority (EFSA) and with other international institutions in these areas, among others [34].

In addition, there are associations that regulate food advertising; in addition to the work of competent health and consumer authorities, the courts, and the advertising industry and the media, such associations include the Association for the Self-Regulation of Commercial Communication (Autocontrol), an independent self-regulator and advertising industry organization in Spain, created in 1995, and the Association of Communication Users (AUC), which is dedicated to the defense of citizen’s interests regarding their relationship with the different media and communication systems and with new information technologies, offering users the possibility of reporting any advertising content they consider illegal [38,39].

The achievement of food advertising must be the work of both advertisers and consumers, who must defend their own rights in accordance with the regulations in force. Specific legislation is needed on the control and quality of nutritional supplements, in which it would be useful to create an official standardized seal for all brands of nutritional supplements in relation to the purity of its composition, similar to those that already exist for gluten-free foods or for organic or biological foods [24].

### 4.4. Cases of Advertising Fraud

Despite the legislation in force and the efforts of organizations such as the European Food Safety Authority (EFSA), the World Anti-Doping Agency (WADA), and the International Olympic Committee (IOC), nutritional supplement marketing fraud is still frequent. This indicates that current legislation needs to be reviewed and strengthened. Athletes are not well informed or advised on the consumption of nutritional supplements; this leads, on the one hand, to the consumption of nutritional supplements without evidence of their benefit or the consumption of substances that can harm health and/or performance due to a lack of information about them. On the other hand, it leads to the purchase and consumption of nutritional supplements that indicate erroneous benefits through incorrect health claims, or that may contain substances prohibited by WADA and are not declared in the nutritional labelling [24].

Studies have shown that the adulteration of food supplements occurs quite frequently, and that most consumers and healthcare professionals are unfamiliar with the problem [40].

In a study on the health claims made in the labeling of substitute drinks, the results showed that none of the claims fully complied with the recommendations, and that 12.3% almost complied with the cause–effect relationship established by the scientific reference documents, which amounts to food fraud towards the consumer [25]. Similarly, a study investigating the health claims associated with creatine monohydrate in commercial marketing found that only 25% of the health claims met the scientific reference criteria. Most of the claims should be modified or removed, as they could be considered fraudulent and/or misleading to the consumer [36]. As in other studies, labeling-related fraud is high, either due to the omission of substances present in the product, or due to errors in the analysis or declaration of ingredient quantities. The lack of truthfulness in the labels and the omission of substances compromise the health and sports performance of the consumer [24].

### 4.5. Limitations of the Study

One of the difficulties of this study was the variability in the results of the search portals, as well as the existence of products that did not offer the information required for the study. This work also highlights the multitude of health claims made by manufacturers or advertisers, which in some cases presented very confusing information. On the other hand, this study focuses on the European context (EFSA scientific opinion) and the scientific evidence and criteria provided by the literature and international institutions. Future research could extend this work. In particular, the European and U.S. perspectives on this issue could be analyzed and compared.

## 5. Conclusions

In summary, according to the research conducted, a total of 60 health claims fully complied (score = 5) with EFSA recommendations. In contrast, 12 health claims received a score of 2, indicating that their wording did not comply with the statements approved by the EFSA. These non-compliant claims represent 5.7% of the total sample (*n* = 209). This means that, although 100% of the products complied with the conditions of use on the label, of all the products analyzed in our study, 43.8% (*n* = 46) of the products named health claims not authorized by the EFSA. In other words, a very large number of SFSs contain information on the label that does not comply with the health claims established by European legislation. This is very worrying in terms of food information and safety for European consumers.

Health claims on proteins must strictly comply with the criteria established by European legislation, consensus documents, and scientific evidence. Food fraud manifests itself in various ways in the advertising, marketing, and commercialization of food, seriously affecting consumers.

This necessarily involves collaboration with consumers, businesses, scientific research, and the European Food Safety Authority to ensure compliance with European regulations and to ensure that manufacturers of this type of supplement are properly informed. Achieving high-quality food advertising is part of the concept of food safety and public health for consumers in relation to this type of product. It is essential to carry out further research that aims to enhance control of the labeling of this type of product and to respond more quickly to these companies, so that the EFSA can act more effectively in the event of unauthorized claims and that the national food safety and nutrition agencies of the various Member States can advise these manufacturers. This will ensure that neither misleading nor confusing information appears on the labels of marketed products.

## Figures and Tables

**Figure 1 nutrients-17-01923-f001:**
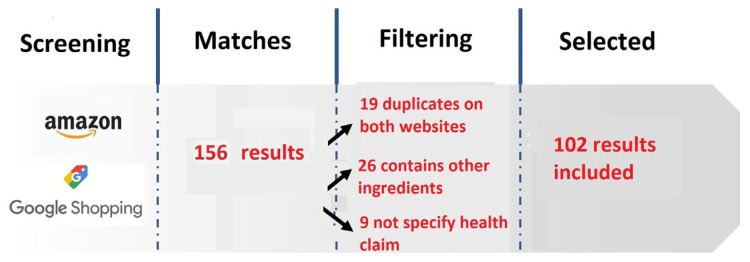
Flow chart showing how the study sample was obtained. Source: Prepared by the authors based on the search data.

**Table 1 nutrients-17-01923-t001:** Effects and applications of protein supplements established by the EFSA’s scientific opinion [28,29].

EFSA	Terms and Conditions of Use	Statement to Be Indicated
Regulation (EC) No. 432/2012 [28] Regulation (EC) No. 1924/2006 [29]	a. Foods that are at least a source of protein: provide at least 12% of the food’s energy value.	Protein contributes to muscle mass increase
Regulation (EC) No. 432/2012 [28] Regulation (EC) No. 1924/2006 [29]	b. Foods that are at least a source of protein: they provide at least 12% of the energy value of the food.	Protein helps to maintain muscle mass
Regulation (EC) No. 432/2012 [28] Regulation (EC) No. 1924/2006 [29]	c. Foods that are at least a source of protein: they provide at least 12% of the energy value of the food.	Proteins contribute to the maintenance of normal bones.
Regulation (EC) No. 1924/2006 [29]	d. If the proteins provide at least 20% of the energy value of the food.	High protein content ^1^

^1^ The content of Table 1 is that contained in the European Register of Health Claims, which details aspects such as characteristics, the claim, the conditions of use of the claims, the relationship with health, reference to the EFSA’s opinion, and legislation. High protein is a nutrition claim, which is any claim that states, suggests or implies that a food has specific beneficial nutritional properties due to its energy, nutrients or other substances (contained or not, or contained in reduced or increased proportions). Only authorized nutrition claims are permitted, provided that they meet the established conditions. Source: Regulation No. 432/2012 and Regulation No. 1924/2006.

**Table 2 nutrients-17-01923-t002:** Distribution of product use conditions according to the health claims and the adequacy of the health claims.

Approved Health Claim	Total Supplements in Which This Statement Is Given		Comply Terms and Conditions of Use *	No. and % Supplements in Which This Dosage Is Given for This Statement	
	NO.	(%)			
Protein contributes to muscle mass increase	99	97.1%	Complies a.	99	1100%
Protein helps to maintain muscle mass	79	77.5%	Complies b.	79	100%
Proteins contribute to the maintenance of normal bones.	31	30.4%	Complies c.	31	100%

* Conditions based on Regulation (EC) No. 432/2012 [28] and Regulation (EC) No. 1924/2006 [29] indicated in Table 1. Note: A sports food supplement may have several nutritional claims.

**Table 3 nutrients-17-01923-t003:** Distribution of health claims not authorized by the EFSA.

Unauthorized Health Claims * Relationship to Health	Health Claims of the Product or Company Website	No. of Supplements Where the Statement Appears	% Supplements Where the Statement Appears
Whey protein: Faster recovery from muscle fatigue after exercise	Promotes muscle recovery	5	7.9%
	Post-workout recovery	7	11.1%
	Improve recovery	4	6.3%
Whey protein: increased endurance capacity during the next bout of exercise after strenuous exercise.	Supports muscle growth for even better performance	4	6.3%
	Objective: Resistance	1	1.6%
	Improved sports performance	4	6.3%
	Improved performance and fast recovery from workouts	1	1.6%
Whey protein: skeletal muscle tissue repair	Whey protein benefits muscle fiber recovery	2	3.2%
	Recommended for those looking to recover their muscle tissues after training.	2	3.2%
	Tissue recovery	2	3.2%
	Key amino acids for proper muscle function and recovery	1	1.6%
Whey protein: growth or maintenance of muscle mass	Whey protein benefits muscle growth	1	1.6%
	Whey protein target muscle development muscle maintenance and recovery	1	1.6%
	Whey protein with amino acids helps increase muscle mass and supports muscle tissue maintenance.	1	1.6%
	Whey protein for muscle volume increase	2	3.2%
Whey protein: increase in muscle strength	Improves strength	1	1.6%
	Increased strength	2	3.2%
Whey protein: reduction of body fat mass during energy restriction and resistance training.	Supports fat loss	1	1.6%
Whey protein: increased satiety leading to reduced energy intake	Helps to reduce appetite, soothe the desire to eat	1	1.6%
Whey protein: growth or maintenance of muscle mass	Stimulate protein synthesis, the process that causes muscles to grow because the amino acids it contains are transported to the muscles through the bloodstream	1	1.6%
	Your protein synthesis is accelerated and your body is able to build muscle.	2	3.2%
Whey protein: increased satiety leading to reduced energy intake	Satiety control	1	1.6%
	Weight loss	1	1.6%
	Helps reduce appetite	2	3.2%
Whey protein: faster recovery from post-exercise muscle fatigue	A premium quality whey protein isolate formulated to feed your muscles fast, so you can recover faster.	1	1.6%
Casein: Growth or maintenance of muscle mass	Perfect micellar casein to prevent catabolization	3	4.8%
	Prevents muscle catabolism	1	1.6%
	Promotes muscle recovery	6	9.5%
	Promotes recovery of damaged fibers against catabolism.	1	1.6%
Hydrolyzed Casein: faster recovery from muscle fatigue after exercise	Supplies the amino acids your body needs to recover while you sleep	1	1.6%

* Unauthorized claims listed in the Community Register (EFSA, 2002) [1]. Note: A sports food supplement may contain several claims. Source: own elaboration based on search data.

**Table 4 nutrients-17-01923-t004:** Distribution of product use conditions according to health claims and their adequacy.

Approved Health Claim	Health Claims of the Product or Company Website	No. of Supplements Where the Statement Appears	% Supplements Where the Statement Appears	Degree of Adequacy *
Protein contributes to muscle mass increase	Protein contributes to muscle mass increase	7	3.3%	Yes
	Protein contributes to the growth and development of muscle mass.	16	7.7%	Yes
	Protein source that helps to increase muscle mass	2	1.0%	No
	Contributes (to the growth, increase, development, gain) of muscle mass	15	7.2%	No
	Protein (100% casein, whey protein, protein content) contributes to the growth, increase of muscle mass.	9	4.3%	No
	(favors, assists, supports, promotes, encourages, perfect for) muscle growth	13	6.2%	No
	Promotes muscle building	1	0.5%	No
	(Helps, favors) muscle gain, muscle mass	4	1.9%	No
	(Helps, favors, promotes) muscle development, muscle mass	21	10.0%	No
	Objective to gain, develop muscle mass	2	1.0%	No
	Designed to help you build muscle	1	0.5%	No
	To increase muscle mass	5	2.4%	No
	Can improve the development of muscle growth	2	1.0%	No
	To accelerate muscle building	1	0.5%	No
Proteins help to preserve muscle mass	Protein helps to maintain muscle mass	6	2.9%	Yes
	Protein contributes to the maintenance of muscle mass	18	8.6%	Yes
	Contribute to the maintenance, maintenance of muscle mass, muscle	6	2.9%	No
	To help maintain, regenerate muscle, muscle mass	21	10.0%	No
	Proteins are important for the maintenance of our muscles.	1	0.5%	No
	Muscle maintenance	8	3.8%	No
	Sustained supply of amino acids to prevent loss of muscle mass	1	0.5%	No
	Protein (100% Whey Isolate, 100% casein, whey protein) contributes to the maintenance of muscle mass.	10	4.8%	No
	Prevents muscle mass/tone loss	1	0.5%	No
	Promotes muscle retention	1	0.5%	No
	To feed your muscles	1	0.5%	No
	For nocturnal muscle support	4	1.9%	No
	Muscle definition	1	0.5%	No
Proteins contribute to the maintenance of normal bones.	Proteins contribute to the maintenance of normal bones.	11	5.3%	Yes
	Source of protein that contributes to the maintenance of bones in a proper state.	2	1.0%	Yes
	100% casein protein contributes to the maintenance of normal bones.	1	0.5%	No
	100% Whey Isolate protein contributes to the maintenance of normal bone structure.	2	1.0%	No
	Protein content contributes to the maintenance of proper bone health	1	0.5%	No
	Contributes to the maintenance of bones in normal conditions.	2	1.0%	No
	Whey protein helps to maintain a normal bone system.	7	3.3%	No
	Maintenance of healthy bones	1	0.5%	No
	Development of your bone health	2	1.0%	No
	Can help you maintain healthy bones	1	0.5%	No
	Promotes bones	1	0.5%	No

* Degree of adequacy if the health claims for the proteins indicated in the selected sample of supplements are adapted to the health claims defined by the EFSA. Source: own elaboration based on search data.

**Table 5 nutrients-17-01923-t005:** Distribution of conditions of use for products according to their nutritional claims and adequacy.

Approved Nutritional Statements	Nutrition Claims Stated on the Product or Company Website	No. of Supplements Where the Statement Appears	% Supplements Where the Statement Appears	Degree of Adequacy ^1^
High protein content	High protein content	2	66.6%	Yes
	Contains high protein	1	33.3%	Yes

^1^ Degree of adequacy if the health claims for the proteins indicated in the selected sample of supplements are adapted to the health claims defined by the EFSA. Source: own elaboration based on search data.

**Table 6 nutrients-17-01923-t006:** Types of non-health-related messages indicated by manufacturers.

Messages Indicated in the Product or Company Website	No. of Supplements Where the Message Appears	% Supplements Where the Message Appears
Increased protein	1	16.7%
Providing the necessary protein value to the diet	1	16.7%
Highly concentrated and balanced whey protein.	1	16.7%
Ideal for increasing your protein intake	2	33.3%
Protein synthesis	1	16.7%

Source: own elaboration based on search data.

**Table 7 nutrients-17-01923-t007:** Summary distribution of the conditions of use (doses, etc.) for the products according to the health claims and their adequacy.

Property Statement Healthy Approved	Total Supplements in Which This Statement Is Given		Comply with Conditions of Use	No. of Supplements Where the Conditions of Use Are Met	% Supplements Where the Conditions of Use for This Statement Are Met	Degree of Adequacy Statement Yes/No	Reason *
	NO.	Total					
Protein contributes to muscle mass increase	99	97.1%	Yes	25	25.3%	Yes	5
			Yes	15	15.2%	No	4
			Yes	58	58.6%	No	3
			Yes	1	1%	No	2
Protein helps to maintain muscle mass	79	77.5%	Yes	24	30.4%	Yes	5
			Yes	27	34.2%	No	4
			Yes	21	26.6%	No	3
			Yes	7	8.9%	No	2
Proteins contribute to the maintenance of normal bones.	31	30.4%	Yes	11	35.5%	Yes	5
			Yes	12	38.7%	No	4
			Yes	4	12.9%	No	3
			Yes	4	12.9	No	2

* Reason according to the EFSA’s scientific opinion: Number 1. Reason: not in accordance with the approved protein claims and the recommended appropriate dosage of the product. Proposed modification: delete product claim. Number 2: Reason: Conforms to the recommended appropriate dosage of the product, but the text of the statement indicated does not conform to the approved one. Proposed modification: modify the declaration by specifying the exact text of the approved declaration Number 3. Reason: Conforms to the appropriate recommended dosage of the product, but the text of the approved statement is missing. Modification proposal: modify the statement by specifying the exact text of the approved statement. Number 4. Reason: Conforms to the appropriate recommended dosage of the product, but some words in the text of the declaration need to be changed. Proposal for amendment: amend the declaration by specifying the exact text of the approved declaration. Number 5. Reason: Conforms to all of the above. Proposed modification: do not modify or delete the declaration. Note: 1 supplement may have several nutrition claims.

## Data Availability

The data presented in this study are available in the tables and Appendix A of this article. The data presented in this study are available upon request from the corresponding authors for privacy reasons.

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
