# Peer review of "Health Claims for Protein Food Supplements for Athletes—The Analysis Is in Accordance with the EFSA’s Scientific Opinion"

_nutrients, 2025, doi:10.3390/nu17111923_

Round 1
Reviewer 1 Report
Comments and Suggestions for Authors The manuscript by deals with a very important topic, which is the reliability of the information provided to consumers in the food supplements label Although it is not the giusto study focused on thisvargument, it is interesting and provide a good overview of the labelling of protein supplements. In this reviewer's opinion, itvshould be suitable for publication inbNutrients after some modifications and clarification. - how was the degree of adequacy evaluated? Criteria should be clearly explained. - Table 1 is a mix of nutrion (high protein) and health claim. Compositional characteristics needed to bear the nutrition claims (source or rich) are the condition of use of health claims. This should be better explained and tanle 1 modified - did AUs evaluate the graphic on the pack or adverts? Imagna are the main source of misleading messages and fraud. Itvshould be considered and discussed - the origin of the products should be indicated. Were all selected supplements produced in the EU? In this case, it would be better to expand the analysis including supplements produced outside the EU. When sold in the EU, they should anyway comply with the EU regulation, but the internet market is sometimes surprising. - Although well written, the introduction is a bit confusing as some sentences refer to fortified foods or to toxic substances or to the presence of not identifies molecules. Since the manuscript focuses on protein supplements and related claims, it is strongly suggested to stick on it - unintentional doping is a problem, but it is due to the presence of specific compounds, and not to fraud in advertising. In this study, no analysis were performed related to doping, so please avoid mentioning - the conclusion section should be completely rewritten. At present, it is a list of results. The conclusion section should emphasize the importance of the results and indicate new research opportunitiesAuthor Response
Reviewer 1
The manuscript by deals with a very important topic, which is the reliability of the information provided to consumers in the food supplements label Although it is not the giusto study focused on thisvargument, it is interesting and provide a good overview of the labelling of protein supplements. In this reviewer's opinion, itvshould be suitable for publication inbNutrients after some modifications and clarification.
- how was the degree of adequacy evaluated? Criteria should be clearly explained.
Response to reviewer:
We have included the reasons for the adequacy of the methodology (Lines 184-202).
- Table 1 is a mix of nutrion (high protein) and health claim. Compositional characteristics needed to bear the nutrition claims (source or rich) are the condition of use of health claims. This should be better explained and tanle 1 modified
Response to reviewer:
“The content of Table 1 is what is available in the European Register of Health Claims, detailing aspects such as characteristics, claim, conditions of use of claims, relationship to health, EFSA opinion reference and legislation.”. This information has been included at the food not of the Table 1.
- did AUs evaluate the graphic on the pack or adverts? Imagna are the main source of misleading messages and fraud. Itvshould be considered and discussed
Response to reviewer:
We have evaluated the product data sheet and the information available on the company's website. It now appears as such throughout the manuscript.
- the origin of the products should be indicated. Were all selected supplements produced in the EU? In this case, it would be better to expand the analysis including supplements produced outside the EU. When sold in the EU, they should anyway comply with the EU regulation, but the internet market is sometimes surprising.
Response to reviewer:
We would like to make your proposal, but this is not possible because European regulations for sports supplements marketed on this continent are being applied. Laws related to advertising, marketing and labelling differ between continents and for that reason it is not possible to apply the same methodology or standards. This has been previously analysed by the authors: https://www.mdpi.com/2072-6643/9/11/1225
- Although well written, the introduction is a bit confusing as some sentences refer to fortified foods or to toxic substances or to the presence of not identifies molecules. Since the manuscript focuses on protein supplements and related claims, it is strongly suggested to stick on it.
Response to reviewer:
Thanks for the suggestion. The introduction has been reduced.
- unintentional doping is a problem, but it is due to the presence of specific compounds, and not to fraud in advertising. In this study, no analysis were performed related to doping, so please avoid mentioning.
Response of the authors: Doping was not the focus of this study, but the authors want to emphasise that it is a type of fraud and that it is related to advertising, the presence of undeclared substances and that it can be tested positive in anti-doping controls. For this reason it has been briefly mentioned in the discussion section, in order to generate discussion on different aspects of consumer fraud in relation to supplements.
.
- the conclusion section should be completely rewritten. At present, it is a list of results. The conclusion section should emphasize the importance of the results and indicate new research opportunities
Response to reviewer:
The conclusions have been completely reworked according to the reviewer's suggestions.

Reviewer 2 Report
Comments and Suggestions for Authors
This is an interesting study looking at the health claims on protein-based sports food supplements.
This manuscript needs significant improvement in all sections. I have attached the file with my comments. The statements have not been expressed clearly, and there needs to be a lot of work done to organise the content better. Some of the phrases are not clear and, therefore, not easy to understand. The use of English and how sentences are phrased need improvement. At some places, the paragraphs are too wordy and this hinders comprehension. There are obvious errors, such as repetition of information in lines 58 to 64.
Visuals (tables) are not easy to understand, most of them lack clear description rows/columns. Some contain errors, such as a column labelled “total” but showing percentages (table 2). Data needs to be presented more efficiently and professionally. There is no need to quote the data in the text without any additional information, ie a brief summary of trends or data that is interesting and therefore stands out from the rest of the dataset.
The discussion and conclusion include numerical results, which should not be the case. The discussion also includes generic information that does not always link to the data obtained in the study.
There is a need to proofread and edit this manuscript.

Author Response
Reviewer 2
This is an interesting study looking at the health claims on protein-based sports food supplements.
This manuscript needs significant improvement in all sections. I have attached the file with my comments. The statements have not been expressed clearly, and there needs to be a lot of work done to organise the content better. Some of the phrases are not clear and, therefore, not easy to understand. The use of English and how sentences are phrased need improvement. At some places, the paragraphs are too wordy and this hinders comprehension. There are obvious errors, such as repetition of information in lines 58 to 64.
Response to reviewer: The manuscript has been improved in all sections. We have also removed repetitions in lines 58-64. The English has been corrected and revised by Deepl.com.
Visuals (tables) are not easy to understand, most of them lack clear description rows/columns. Some contain errors, such as a column labelled “total” but showing percentages (table 2). Data needs to be presented more efficiently and professionally. There is no need to quote the data in the text without any additional information, ie a brief summary of trends or data that is interesting and therefore stands out from the rest of the dataset.
Response to reviewer: The tables have been improved to make the data presented more understandable.
The discussion and conclusion include numerical results, which should not be the case. The discussion also includes generic information that does not always link to the data obtained in the study.
Response to reviewer: the discussion and conclusions have been improved.
There is a need to proofread and edit this manuscript.

Reviewer 3 Report
Comments and Suggestions for Authors
Dear authors,
The abstract is very well done. I would suggest, however, to insert at the end a sentence about the implications of the study (theoretical, managerial, practical, etc).
The introduction and the literature review contain a lot of information, but not enough to see a red thread and to understand what the research problem is. The introduction starts by presenting the general context, but then it should continue with the identification of the gap in the literature, the presentation of the problem researched, and finally the presentation of how the paper is structured.
The methodology part needs to be improved. Firstly, I would suggest not to use bullets and numbering in a scientific article. The variables studied for each product could be presented either as a table or as a figure. Same remark for data analysis. Also, in this section, the authors do not present enough information so that the research can be replicated. How was this content analysis done? Was a particular software used?
The results and discussions are nicely presented, but the conclusions should not repeat what has already been said. Also in this section, the authors should present the theoretical implications of the study, but also the practical ones (managerial, social, etc). The authors should also present future research directions.
Author Response
Reviewer 3
The abstract is very well done. I would suggest, however, to insert at the end a sentence about the implications of the study (theoretical, managerial, practical, etc).
Response to reviewer:
The sentence has been included.
The introduction and the literature review contain a lot of information, but not enough to see a red thread and to understand what the research problem is. The introduction starts by presenting the general context, but then it should continue with the identification of the gap in the literature, the presentation of the problem researched, and finally the presentation of how the paper is structured.
Response to reviewer:
the introduction has been additionally completed.
The methodology part needs to be improved. Firstly, I would suggest not to use bullets and numbering in a scientific article. The variables studied for each product could be presented either as a table or as a figure. Same remark for data analysis. Also, in this section, the authors do not present enough information so that the research can be replicated. How was this content analysis done? Was a particular software used?
Response of the authors: It is a methodology already used in 3 previously published articles: doi:10.1017/S1368980020005121. doi:10.3390/molecules26072095. doi:10.3390/nu16131980. The information concerning to the section 2.5 has been enlarged to better explaining all the questions addressed by the reviewer. No particular software has been used as it does not exist.
The results and discussions are nicely presented, but the conclusions should not repeat what has already been said. Also in this section, the authors should present the theoretical implications of the study, but also the practical ones (managerial, social, etc). The authors should also present future research directions.
Response to reviewer:
The conclusions have been completely reworked according to the reviewer's suggestions.
Round 2
Reviewer 1 Report
Comments and Suggestions for Authors
The revised manuscript has been significantly improved and all the reviewer's concerns have been adequately addressed and considered. In the opinion of this reviewer, the manuscript is now acceptable for publication in Nutrients, but a small revision is still required. Indeed, the wording of approved health claims is not set in stone and can be changed upon request and approval by EFSA, unless it changes the meaning. Therefore, there are claims on some products that do not appear to be 100% compliant with the approved health claim, but the change could have been approved by EFSA. Have the authors checked this possibility? If not, the issue should be included in the discussion.
Author Response
In line with the editor's comments, we would like to clarify that we relied on claims approved by the European Union Commission and available in the European Register of Health Claims (website: https://ec.europa.eu/food/food-feed-portal/screen/health-claims/eu-register).
To clarify this information, this information has been added to section 2.6 and Table 1 of the Materials and Methods section.
